# Surgical Treatment of Cystic Pituitary Prolactin-Secreting Macroadenomas: A Single Center Study of 42 Patients

**DOI:** 10.3390/brainsci12060699

**Published:** 2022-05-27

**Authors:** Xiang Guo, Juan Chen, Zhuo Zhang, Xueyan Wan, Kai Shu, Ting Lei

**Affiliations:** Department of Neurosurgery, Tongji Hospital, Tongji Medical College, Huazhong University of Science and Technology, Wuhan 430030, China; d202082008@hust.edu.cn (X.G.); jchen@tjh.tjmu.edu.cn (J.C.); zhzmclaren@gmail.com (Z.Z.); xywan@tjh.tjmu.edu.cn (X.W.); kshu@tjh.tjmu.edu.cn (K.S.)

**Keywords:** cystic pituitary prolactin-secreting macroadenomas, extra-pseudocapsular transsphenoidal surgery, bromocriptine

## Abstract

This study evaluated the therapeutic effects of surgical treatment of cystic pituitary prolactin-secreting macroadenomas. The clinical data of 42 patients with cystic pituitary prolactin-secreting macroadenomas were retrospectively analyzed. Patients were divided into medication plus surgery and surgery alone groups based on the regularity of bromocriptine treatment before surgery. Both groups underwent extra-pseudocapsular transsphenoidal surgery for tumor resection, and postoperative images and clinical follow-up were retrospectively reviewed. We also evaluated patients who opted for long-term treatment with bromocriptine. In the medication plus surgery group, the long-term surgical cure rate and comprehensive remission rate were 33.3% and 41.7%, while in the surgery alone group they were 69.2% and 80.8%, respectively. No severe or permanent complications occurred, and the surgical complication morbidity rate was 10.5%. The rate of tumor progression during the long-term follow-up was 33.3% and 7.7% in the medication plus surgery and surgery alone groups, respectively. The time required for prolactin levels to return to normal in the surgery alone group was significantly faster and the proportion that returned to normal was significantly higher. Direct surgical treatment after diagnosis combined with postoperative individualized bromocriptine adjuvant therapy had better efficacy in patients with cystic pituitary prolactin-secreting macroadenomas, but its long-term effectiveness requires further follow-up.

## 1. Introduction

Pituitary prolactin-secreting adenomas are the most common functional pituitary adenomas, accounting for approximately 25–41% of pituitary adenomas and 50–60% of functional pituitary adenomas [1,2]. Cystic pituitary prolactin-secreting adenomas contain cystic parts that account for more than 50% of the total tumor volume [3]. Currently, dopamine agonists (DAs) are highly effective in normalizing prolactin (PRL) and ameliorating the symptoms of hyperprolactinemia [4]. The preferred treatment for prolactinomas is medical therapy; in contrast, transsphenoidal surgery (TSS) is only used as a second-line treatment [5]. However, cystic prolactinomas respond poorly to DAs owing to a lack of dopamine receptors in the cystic part and often require surgical treatment [3]. Thus, the treatment strategy and timing of surgical intervention for cystic pituitary prolactin-secreting adenomas remains controversial. 

The therapeutic approach for cystic prolactinomas has not been fully described in clinical guidelines [6]. Only a few cases have reported treatment outcomes. Most cystic pituitary prolactin-secreting adenomas are large in size [3]. Therefore, the purpose of this study was to confirm the efficacy of DAs and surgical intervention as clinical therapy for cystic pituitary prolactin-secreting macroadenomas. Most importantly, we comprehensively evaluated the efficacy of different treatments and elucidated the possible indication for a shift to primary transsphenoidal adenomectomy for cystic pituitary prolactin-secreting macroadenomas as the primary therapy.

## 2. Materials and Methods

### 2.1. Patients

We prospectively enrolled 42 consecutive patients with cystic pituitary prolactin-secreting macroadenomas that were treated between January 2016 and May 2019. Sixteen received regular bromocriptine therapy and 12 eventually underwent surgery. Routine follow-ups continued until the end of December 2021 by phone or in an outpatient clinic. This study obtained ethical approval from the Research Ethics Committee of the Huazhong University of Science and Technology, People’s Republic of China. Informed consent was obtained from all patients.

The inclusion criteria were as follows: (1) clinical symptoms: men with erectile dysfunction and sexual dysfunction or women with menstrual disorders, amenorrhea, lactation, infertility, and other manifestations. (2) PRL level: the Clinical Laboratory of Tongji Hospital defined a normal PRL level as 2.64–13.13 ng/mL in men, 3.34–26.72 ng/mL in women before menopause, and 2.74–19.64 ng/mL after menopause. A PRL level significantly higher than the normal upper limit was defined as hyperprolactinemia. PRL level of <200 ng/mL, as in conditions such as intake of various drugs, compression of the pituitary stalk by other pathologies, hypophysitis, or idiopathic hyperprolactinemia, need to be considered. (3) Magnetic resonance imaging (MRI): pituitary tumor >1 cm in diameter, containing a cystic component that accounts for at least 30% of the tumor volume and with distinguishable non-adenomatous lesions. (4) Histopathology: Prolactinomas diagnosed by postoperative pathological examination, and single PRL positivity detected by immunohistochemistry. Cases of multiple hormone-secreting adenomas (such as PRL and GH, or PRL, and TSH) were excluded.

In our study, a total of 42 patients were enrolled, of whom 16 patients preferred medical therapy and 26 preferred surgery. Of the 16 patients treated with the preferred medical therapy, four received long-term medical therapy, and 12 eventually underwent surgery.

### 2.2. Pharmacotherapy

All 12 patients in the medication plus surgery group were treated with bromocriptine at a therapeutic dose of 2.5–15 mg/d. During treatment, the dose was gradually adjusted according to their PRL levels and any tumor changes identified by MRI. One patient discontinued bromocriptine one month after the onset of treatment owing to adverse effects such as dizziness and vomiting. Two patients developed cerebrospinal fluid rhinorrhea three months after starting therapy and were immediately hospitalized and needed surgery. The remaining patients reached a standard treatment course of six months. Moreover, four patients were treated with bromocriptine on a long-term basis at an initial dose of 10–15 mg/d, and in three patients the dose was gradually reduced to 2.5–7.5 mg/d after normal PRL levels were achieved.

### 2.3. Surgery Method

All patients underwent trans-septal-sphenoidal approach to the sellar and extra-pseudocapsular tumor resection assisted by neuro-navigation and neuro-endoscopy, mainly performed by the senior author (L.T.), who is an experienced pituitary surgeon with experience of performing over 300 transsphenoidal surgeries (TSSs) per year. A microdissector was employed to separate the border at the interface between the pituitary gland and the tumor. To remove the tumor en bloc, the pseudocapsule plane from the pituitary side was gently detached and preserved, followed by the surrounding part. For macroadenomas, sufficient intratumoral decompression was necessary prior to extracapsular dissection. If no definite pseudocapsule could be found, or the pseudocapsule was discontinuous, the tumor was excised in pieces.

### 2.4. Remission and Tumor Progression Criteria

During the perioperative period, pituitary function was closely evaluated and monitored using symptoms, signs, and endocrine examination. Outpatient follow-up was performed at one month, three months, and yearly thereafter. MRI examinations were performed before and three months after surgery. Follow-up MRIs were performed annually.

Initial remission was considered to have occurred if the PRL levels were within the normal range on the first postoperative day. Twelve weeks post-surgery was the earliest time point for the evaluation of surgical remission, which was achieved when the patient’s symptoms had resolved, the tumor mass had disappeared (certified by gross total resection on MRI), and PRL levels were normal.

Tumor progression was defined when MRI showed tumor recurrence after gross total resection (GTR), increased volume of residual tumor that was not GTR, or there was a sustained increase in PRL levels. Long-term remission was defined at the latest follow-up before December 2021, as the absence of hyperprolactinemia or DAs medication.

### 2.5. Statistical Analysis

We used descriptive statistics, contingency coefficients (CCs), and standardized mean differences (SMDs) to indicate the sample size-independent magnitude of group differences. Pearson’s chi-square test, Fisher’s exact test, two-sample *t*-test, or Mann–Whitney U-test were used depending on the scaling and distribution of the variables. The time required for serum PRL levels to return to normal was estimated using the Kaplan–Meier method. Statistical significance was set at *p* < 0.05. All statistical analyses were performed using SPSS Statistics version 21 (IBM SPSS Statistics Inc., Chicago, IL, USA).

## 3. Results

### 3.1. Clinical Characteristics

Forty-two patients were eligible for analysis: 20 men and 22 women. Four male patients were treated with bromocriptine for a long time period. They had a mean age of 42.5 ± 9.8 years (range: 31–54 years), and the initial mean PRL level before bromocriptine treatment was 1152.2 ± 725.8 ng/mL (range: 502.3–2137.4 ng/mL). In 38 surgical patients, the mean age was 38.6 ± 13.5 years (range: 16–63 years), and the mean preoperative PRL level was 1169.9 ± 1496.9 ng/mL (range: 1.6–5140.8 ng/mL). Furthermore, 22 patients (61.1%) had PRL levels greater than 200 ng/mL. DAs (bromocriptine (the only CFDA-permitted prescribed DA in China)) were primarily used in 16 patients (38.1%). Nine cases (23.7%) had a postoperative tumor pathology showing a Ki-67 index of ≥3%. The results are shown in Table 1 and Table 2.

### 3.2. The Therapeutic Effects of Bromocriptine

Sixteen patients (12 men and four women) were initially treated with bromocriptine after the discovery of cystic pituitary prolactin-secreting macroadenomas. Four male patients were treated with bromocriptine for a long time period. At the final follow-up, PRL levels were normal in three patients, and tumor signals with cystic lesions were visible on MRI in two patients (Table 3).

For the other 12 patients, the mean tumor diameter before bromocriptine was 3.3 ± 1.7 cm (range: 1.5–6.8 cm), the mean cyst diameter was 1.4 ± 1.0 cm (range: 0.4–2.1 cm), and the mean PRL level was 3076.5 ± 2514.7 ng/mL (range: 275.4–8000 ng/mL). They were treated with bromocriptine at a dose of 9.8 ± 4.8 mg/d (range: 2.5–15 mg/d) for a mean duration of 9.8 ± 7.7 months (range: 1–24 months). After bromocriptine treatment, the mean tumor diameter was reduced to 2.8 ± 1.7 cm (range: 0.9–6.6 cm), the mean cyst diameter was reduced to 1.2 ± 0.8 cm (range: 0.3–1.8 cm), and the mean PRL level was reduced to 1496.1 ± 1614.4 ng/mL (range: 1.55–5140.78 ng/mL). Only one patient developed severe vomiting during drug treatment, which resulted in discontinuation after one month. Eight patients had a poor effectiveness of bromocriptine treatment; of these, two patients had an ideal reduction in PRL levels after treatment; however, their tumor size did not decrease significantly, and they still had clinical manifestations, such as headaches and vision loss, caused by tumor compression. In addition, two patients developed cerebrospinal fluid (CSF) rhinorrhea and one patient developed tumor stroke. Table 4 and Table 5 describe the 12 patients that were treated with extra-pseudocapsular transsphenoidal surgery (EPTSS).

### 3.3. Surgical Remission and Postoperative Complications

The 38 patients who underwent surgical treatment were divided into the plus surgery and surgery alone groups (12 and 26, respectively). There were no statistically significant differences in sex, age, tumor size, number and size of cystic lesions, and Knosp grade between the two treatment groups (all were *p* > 0.05), and their baselines was comparable. In the medication plus surgery group, the median postoperative PRL level was 224.7 ng/mL (range: 1.7–4534.5 ng/mL) on the first day after surgery, and the mean postoperative PRL levels were 859.8 ± 1362.0 ng/mL and 1020.8 ± 2256.9 ng/mL on the first day and at the 12th week, respectively. Only three patients (25%) underwent total tumor resection. At the 12th week of multidimensional evaluation, only four patients (33.3%) achieved surgical remission. In comparison, in the surgery alone group, the median postoperative PRL level was 27.5 ng/mL (extreme values: 1.5–610.8 ng/mL) on the first day after surgery, and the mean postoperative PRL levels were 98.7 ± 153.3 ng/mL and 80.6 ± 128.1 ng/mL on the first day and 12th week, respectively. Total tumor resection was achieved in 13 (50%) patients. At the 12th week assessment, 18 patients (69.2%) had achieved surgical remission (Table 6).

For the 16 patients who did not meet the surgical remission criteria, DAs therapy was immediately initiated for those with hyperprolactinemia. Most patients experienced symptom relief and remission, except for one male patient who did not respond to bromocriptine postoperatively.

No perioperative mortality or major morbidity was observed in this cohort. Four patients developed different types and degrees of postoperative complications, and the overall incidence rate was 10.5%. One patient (2.6%) experienced epistaxis and required emergency tamponade. Two patients (5.3%) had CSF rhinorrhea in the hospital which was relieved after lumbar cistern drainage. Within the first week post-surgery, three patients (7.9%) had transient diabetes insipidus, two required medication management, and none required additional intervention after discharge. No cases of syndrome of inappropriate antidiuretic hormone (SIADH) were observed. Two patients had postoperative anterior pituitary hypofunction (1 case was thyroidal axis, 1 case was adrenal axis) but no special discomfort; therefore, they did not receive hormone replacement therapy. Their anterior pituitary function gradually returned to normal during the two reexaminations one and three months after surgery (Table 7).

Long-term follow-up data ranging from 25 to 76 months were available (51.0 months on average) for 42 patients. Seven patients (four in the medication plus surgery group, two in the surgery alone group, and one in the medication alone group) had confirmed hyperprolactinemia or tumor progression after a median period of 8.5 months (range: 3–20 months). Only three patients (18.8%) achieved long-term remission with bromocriptine treatment. At the final follow-up, five patients (41.7%) in the medication plus surgery group had achieved long-term remission and were defined as being surgically cured. In the surgery alone group, a total of 21 patients (80.8%) achieved a surgical cure (Table 7).

Only one patient in the medication plus surgery group was in remission after long-term treatment with DAs. However, in the surgery alone group, two patients successfully ceased bromocriptine after 1.5 years, and one patient was in remission after long-term treatment with DAs. Two individuals with a long history of treatment with DAs eventually developed resistance, and subsequently underwent surgery.

Kaplan–Meier analysis was used to compare the time required for PRL levels to return to normal between the two treatment groups. We found that the time required for the PRL level to return to normal was significantly shorter in the surgery alone group than that in the medication plus surgery group, and the proportion of PRL levels to return to normal was significantly higher as shown in Figure 1. 

## 4. Discussion

Pituitary prolactin-secreting adenomas are the most common functional pituitary adenomas, accounting for approximately 50% of all pituitary adenomas that require treatment [6]. Their clinical symptoms can be divided into two categories: first, excessive PRL secreted by tumors leading to sexual dysfunction in men and menstrual disorders including amenorrhea, infertility, lactation, and other symptoms in women; and second, mass compression effect caused by the large tumor size, leading to headache, nausea and vomiting, visual impairment, and cranial nerve dysfunction [2]. Cystic pituitary prolactin-secreting adenomas are a special type of pituitary prolactin-secreting adenoma, and some studies suggest that their occurrence may be related to spontaneous hemorrhage and necrosis of the adenomas, or caused by the combined action of radiotherapy, traumatic stress, and anticoagulant use [7]. DAs are currently the preferred and main line of treatment for almost all pituitary prolactin adenomas. Bromocriptine is mainly used in China, whereas Das, such as cabergoline and quinolide, are mainly used in other countries [4]. In this study, 12 patients in the medication plus surgery group were first treated with bromocriptine, at a dose of 2.5–15 mg/d, with a mean of 9.8 ± 4.8 mg/d. The treatment course lasted between 1 and 24 months, with a mean of 9.8 ± 7.7 months. However, our results showed poor efficacy, and the patients eventually underwent surgery for drug intolerance, drug resistance, CSF rhinorrhea, and tumor stroke. Only four patients were treated with bromocriptine for a long period of time, of which three achieved remission. However, two patients still had cystic lesions on MRI. Studies have suggested that the cystic part of pituitary prolactin-secreting adenomas lacks dopamine receptors compared to that in the solid part, contributing to the poor therapeutic effect and often resulting in the need for surgical intervention [3]. It has also been suggested that DAs should be the first choice of treatment for cystic pituitary prolactin-secreting adenomas [3,8]. However, due to the lack of systematic and substantial clinical data, an effective treatment plan outlining the timing for surgical treatment remains to be further explored.

Cystic prolactinomas tend to occur in young patients with an age of onset of 30–40 years [9]. The mean age of onset of the patients in our study was 38.9 years, which was consistent with that in previous reports. Our sample mostly consisted of men, possibly because male pituitary prolactin-secreting adenomas are mostly macroadenomas, and macroadenomas are often accompanied by cystic changes [3]. However, the incidence of pituitary prolactin-secreting adenomas in women was three times that in men, and the overall incidence of cystic pituitary prolactin-secreting macroadenomas was higher in women [10,11]. For most patients with prolactinomas, regular use of DAs can significantly reduce tumor volume and PRL levels, and significantly improve the symptoms of hyperprolactinemia and tumor mass compression [4,12]. However, we found that when cystic pituitary prolactin-secreting macroadenomas were reduced to a certain extent, even if the maximum safe dose of bromocriptine treatment was maintained, the tumor did not continue to shrink, and the PRL levels of the patients could not be reduced to a normal level. In the medication plus surgery group, the mean tumor diameters of the 12 patients were reduced from 3.3 ± 1.7 to 2.8 ± 1.7 cm after regular bromocriptine treatment. The mean PRL level decreased from 3076.5 ± 2514.7 to 1496.1 ± 1614.4 ng/mL; however, this difference was not statistically significant. Among the three patients who achieved long-term remission with bromocriptine, MRI still showed cystic lesions in one patient and solid tumor remnants in another. Consistent with the poor efficacy of bromocriptine treatment, the PRL levels of the patients did not return to normal with endocrinological symptoms, and some patients still showed symptoms of tumor mass compression. Ultimately, the 12 patients who received regular bromocriptine treatment underwent surgical treatment. Although guidelines still list DAs as the preferred treatment method for prolactinomas, the indications for surgical treatment have gradually expanded in recent years. In addition to the classical indications for acute complications (tumor stroke or CSF rhinorrhea), drug intolerance or poor drug effects, tumor enlargement, and symptom aggravation during pregnancy, surgery is also a very important treatment option for young women who have not given birth, patients who are unwilling to accept long-term drug therapy, and patients with excessive drug therapy [13]. EPTSS for tumor resection minimizes tumor compression and reduces the PRL levels of patients, but also increases the therapeutic effect of DAs. A study that reviewed the efficacy of a series of medications in patients with cystic prolactinomas showed that cabergoline (or cabergoline in combination with bromocriptine) reduced tumor size in approximately 70% of patients [3]. In the largest series of reported cases, cabergoline-treated patients had a median cyst reduction rate of 83.5% over a median of 24.6 weeks; however, 50% of patients still underwent surgery for a variety of reasons [9]. These results also suggest that, for cystic prolactinomas, only a small proportion of the population may be able to receive long-term effective treatment with DAs. In addition, few cases of bromocriptine therapy have been reported, and its efficacy remains unclear. Therefore, all patients were directly treated surgically later in our study, and some patients were individually treated with bromocriptine after surgery to determine surgical timing and an effective treatment plan for cystic pituitary prolactin-secreting macroadenomas.

Since we mainly studied macroadenomas, laboratory testing errors and the existence of pituitary stalk effect should be considered comprehensively as shown by a retrospective analysis of PRL levels [14]. Compared with that in the medication plus surgery group, patients in the surgery alone group had a higher rate of initial remission, but this difference was not statistically significant. However, at the follow-up of three months after surgery, the surgical remission rate in the surgery alone group was significantly higher than that in the medication plus surgery group, indicating that regular bromocriptine treatment before surgery did not benefit subsequent surgical treatment. The remission rate was positively correlated with the degree of tumor resection, and the GTR was low in the medication plus surgery group, which may be related to the fact that the patients regularly used bromocriptine before surgery to cause tumor fibrosis, thus increasing the complexity of the surgery [15].

Most cystic prolactinomas are macroadenomas with a high Knosp grading and low surgical GTR. If postoperative PRL levels fail to return to normal, individualized treatment with DAs and regular follow-up are required [16]. Some studies have suggested that patients with cystic prolactinomas should be treated with DAs even if they have symptoms of optic nerve compression, and surgical treatment should be considered only when there are poor responses to DAs and acute complications [8,16,17]. However, some studies also found that surgical treatment had a better remission rate as a potential therapeutic alternative for cystic pituitary prolactin-secreting macroadenomas [18]. In terms of the surgical efficacy of prolactinomas, Gillam reported that the remission rates of surgical treatment for microadenomas and macroadenomas were 75% and 34%, respectively [19]. However, remission rates reported by Chanson and Maiter were higher (81% for microadenomas and 41% for large adenomas). Surgical efficacy varies greatly in prolactinomas, with remission rates ranging from 60 to 93% for microadenomas and from 10 to 74% for large adenomas [20]. At the final follow-up of this study, the long-term remission rate of the surgery alone group was 80.8%, and the proportion of patients treated with bromocriptine after surgery was 30.8%. In the medication plus surgery group the two rates were 41.7% and 66.7%, respectively. The postoperative tumor progression rate was lower in the surgery alone group than that in the medication plus surgery group. Although there was no statistical difference, this may be related to the small sample size, and further expansion of the sample size may yield meaningful results in the future. We also used a Kaplan–Meier analysis and found that it took significantly less time for PRL levels to return to normal in the surgery alone group, indicating that these patients had a higher long-term remission rate and a shorter treatment period, which is of great significance. Cystic pituitary prolactin-secreting macroadenomas occur mainly in young patients. Shortening the treatment time of the disease and restoring the PRL level to normal faster can result in a faster disappearance of clinical symptoms and allow the patients to return to normal social life.

In cystic prolactinomas, surgery can increase the sensitivity to DAs therapy and improve conditions for postoperative radiation therapy [21,22,23,24]. At present, some studies have had different views on the effect of preoperative DAs treatment on surgery, which remain uncertain [25,26,27]. Based on our results, we propose that patients with cystic pituitary prolactin-secreting macroadenomas should be treated directly with surgical intervention, followed by an individualized postoperative adjuvant DAs treatment plan. This would lead to a better prognosis than that with the regular treatment plan of DAs until drug tolerance or acute complications. Surgical treatment should be considered the first choice of therapy for patients with cystic pituitary prolactin-secreting macroadenomas.

This study has several limitations. First, a relatively small number of participants were included because this surgery is not routinely performed in patients with prolactinomas. Second, cabergoline is not currently available on the Chinese mainland; therefore, bromocriptine was used as the only DA in this study. Thus, we cannot comment on the efficacy of cabergoline in the treatment of cystic pituitary prolactin-secreting macroadenomas. Third, this was an observational study, and we only focused on the surgical outcomes at a single center. Additionally, the study has inherent limitations owing to its retrospective nature. Therefore, we must emphasize that these findings cannot be universally applied to all neurosurgical institutes. Importantly, all patients in this cohort were treated in a high-volume pituitary facility by a highly skilled pituitary surgeon who performs over 300 pituitary surgeries annually. Furthermore, the specific indications for surgery may have introduced a bias in the selection of patients studied. The impacts of this (positive or negative) on the reported outcomes are not clear. Currently, the different therapeutic techniques used in each individual case should be the outcome of a multidisciplinary discussion between experienced endocrinologists and pituitary surgeons. Additional well-designed case-control studies should be conducted to confirm the efficacy of transsphenoidal surgery in conjunction with medical therapy for cystic prolactinomas.

## 5. Conclusions

Cystic prolactinomas are a unique form of prolactinoma with no clear standard of care in the current clinical guidelines. Our data suggest that cystic pituitary prolactin-secreting macroadenomas respond poorly to bromocriptine and may ultimately require surgical treatment. For cystic pituitary prolactin-secreting macroadenomas, direct surgical treatment combined with individualized postoperative DAs therapy may be more beneficial to patients. Although this is not the preferred treatment for prolactinomas, this may be useful to neurosurgery clinicians treating these disorders. However, further study is needed to support this conclusion.

## Figures and Tables

**Figure 1 brainsci-12-00699-f001:**
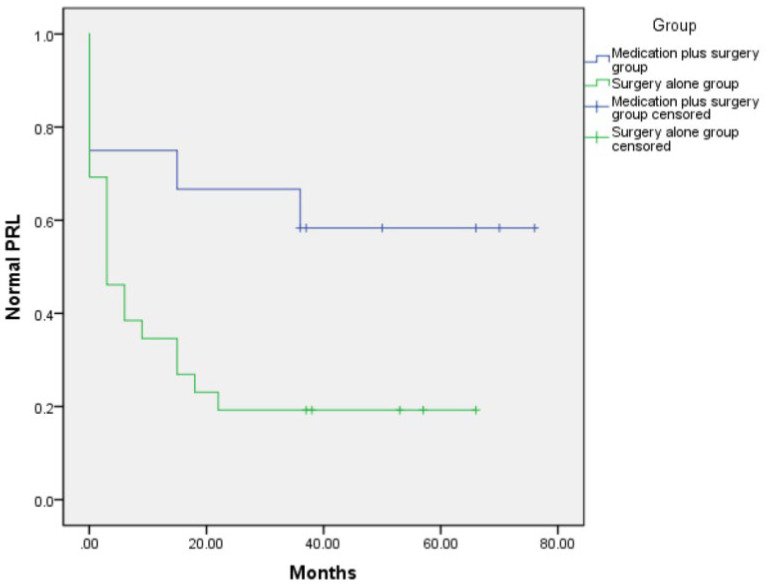
Kaplan–Meier analysis of postoperative time required for PRL level to return to normal in 38 patients with cystic pituitary prolactin-secreting macroadenomas, according to the different treatment plan. They were divided into the medication plus surgery and surgery alone groups.

**Table 1 brainsci-12-00699-t001:** Baseline characteristics of 38 patients with surgically treated cystic pituitary prolactin-secreting macroadenomas.

Variables	Patients (*n* = 38)
Age, years (mean ± SD)	38.6 ± 13.5
Gender, male/female (%)	16 (42.1)/22 (57.9)
Tumor diameter, cm (mean ± SD)	2.6 ± 1.3
Ki-67 (%)	
<3	29 (76.3)
≥3	9 (23.7)
Follow-up time, months (median)	52.5
Pre-operation PRL level, ng/mL (mean ± SD)	1169.9 ± 1496.9
Clinical symptoms (%)	
Headache	19 (50.0)
Diminution of vision	13 (34.2)
Menstrual disorder	16 (42.1)
Galactosis	4 (10.5)
Hyposexuality	3 (7.9)

**Table 2 brainsci-12-00699-t002:** Clinical characteristics of 42 patients of cystic pituitary prolactin-secreting macroadenomas with different treatment methods.

	Medication Alone Group	Medication Plus Surgery Group	Surgery Alone Group	*p* Value
Total (*n*)	4	12	26	
Age, years (mean ± SD)	42.5 ± 9.8	35.2 ± 12.1	40.1 ± 14.0	0.486 ^1^
Gender (male/female)	4/0	8/4	8/18	0.008 ^2^
Tumor diameter, cm (mean ± SD)	3.4 ± 1.3	3.3 ± 1.7	2.3 ± 1.0	0.220 ^1^
Number of cysts (mean ± SD)	1.3 ± 0.5	1.3 ± 0.5	1.3 ± 0.6	0.871 ^1^
Cysts diameter, cm (mean ± SD)	2.0 ± 0.7	1.4 ± 1.0	1.0 ± 0.6	0.020 ^1^
PRL before treatment (ng/mL)	1152.19 ± 725.84	1609.61 ± 1689.63	966.89 ± 1387.21	0.450 ^1^
Knosp grade (%)				0.480 ^3^
Grade 0	0 0)	0 (0)	1 (3.8)	
Grade I	0 (0)	2 (16.7)	3 (11.5)	
Grade II	1 (25.0)	2 (16.7)	4 (15.4)	
Grade III	1 (25.0)	1 (8.3)	11 (42.3)	
Grade IV	2 (50.0)	7 (58.3)	7 (26.9)	
Follow-up time, months (mean ± SD)	51.5 ± 18.4	55.3 ± 13.7	48.9 ± 11.4	0.368 ^1^
Number of patients with long-term DAs	3 (75.0)	7 (58.3)	5 (19.2)	0.014 ^2^

^1^ is one-way ANOVA, ^2^ is Fisher’s exact test, ^3^ is nonparametric Kruskal–Wallis test.

**Table 3 brainsci-12-00699-t003:** Clinical characteristics and outcomes of 4 patients treated with bromocriptine for a long time.

	Case 1	Case 2	Case 3	Case 4
Gender (M/F)	M	M	M	M
Age, years	39	31	54	46
Tumor diameter before bromocriptine treatment, cm	2.5	4.9	4	2
Number of cysts before bromocriptine treatment	1	2	1	1
Cysts diameter before bromocriptine treatment, cm	1.5	2.5, 2.3	2.7	1.1
PRL before bromocriptine treatment, ng/mL	2137.4	502.32	728.25	1240.78
Bromocriptine therapeutic dose, mg/d	10	15	10	15
Follow-up time, months	65	63	53	25
Tumor diameter after bromocriptine treatment, cm	0.4	0.1	0.8	1.5
Number of cysts after bromocriptine treatment	0	0	1	1
Cysts diameter after bromocriptine treatment, cm	0	0	0.8	0.4
PRL at last follow-up, ng/mL	0.53	15.38	26.39	232.90

**Table 4 brainsci-12-00699-t004:** The curative effect for 12 patients of operation treated regularly treated with bromocriptine.

	Before Bromocriptine	After Bromocriptine	*p* Value
Bromocriptine dose, mg/d (mean ± SD)	9.8 ± 4.8		
Bromocriptine use time, months (mean ± SD)	9.8 ± 7.7		
Tumor diameter, cm (mean ± SD)	3.3 ± 1.7	2.8 ± 1.7	0.480 ^1^
Cyst diameter, cm (mean ± SD)	1.4 ± 1.0	1.2 ± 0.8	0.506 ^1^
PRL level, ng/mL (mean ± SD)	3076.5 ± 2514.7	1496.1 ± 1614.4	0.081 ^1^

^1^ is *t* test.

**Table 5 brainsci-12-00699-t005:** Surgical indications for 12 patients regularly treated with bromocriptine.

Surgical Indication	Patients (*n* = 12)
Drug intolerant (%)	1 (8.3)
Drug resistance (%)	8 (66.7)
CSF rhinorrhea (%)	2 (16.7)
Tumor stroke (%)	1 (8.3)

**Table 6 brainsci-12-00699-t006:** Baseline characteristics of two different treatment groups with cystic pituitary prolactin-secreting macroadenomas.

	Total	Medication Plus Surgery Group	Surgery Alone Group	*p* Value
Total (*n*)	38	12	26	
Age, years (mean ± SD)	38.6 ± 13.5	35.2 ± 12.1	40.1 ± 14.0	0.300 ^1^
Tumor diameter, cm (mean ± SD)	2.6 ± 1.3	3.3 ± 1.7	2.3 ± 1.0	0.081^1^
Number of cysts (mean ± SD)	1.3 ± 0.6	1.3 ± 0.5	1.3 ± 0.6	0.638 ^1^
Cysts diameter, cm (mean ± SD)	1.1 ± 0.7	1.4 ± 1.0	1.0 ± 0.6	0.165 ^1^
Ki-67 (%)				1.000 ^4^
<3	29 (76.3)	9 (75.0)	20 (76.9)	
≥3	9 (23.7)	3 (25.0)	6 (23.1)	
Pre-operation PRL (ng/mL)				
<200	22 (57.9)	9 (75.0)	13 (50.0)	0.147 ^2^
≥200	16 (42.1)	3 (25.0)	13 (50.0)	
Knosp grade (%)				0.293 ^3^
Grade 0	1 (2.6)	0 (0)	1 (3.8)	
Grade I	5 (13.2)	2 (16.7)	3 (11.5)	
Grade II	6 (15.8)	2 (16.7)	4 (15.4)	
Grade III	12 (31.6)	1 (8.3)	11 (42.3)	
Grade IV	14 (36.8)	7 (58.3)	7 (26.9)	
Follow-up time, months (mean ± SD)	50.9 ± 12.4	55.3 ± 13.7	48.9 ± 11.4	0.143 ^1^
Initial remission rate (%)	16 (42.1)	3 (25.0)	13 (50.0)	0.147 ^2^
Surgical remission rate (%)	22 (57.9)	4 (33.3)	18 (69.2)	0.037 ^2^
Long-term remission rate (%)	26 (68.4)	5 (41.7)	21 (80.8)	0.026 ^4^
Tumor progression rate (%)	6 (15.8)	4 (33.3)	2 (7.7)	0.066 ^4^

^1^ is *t* test, ^2^ is χ^2^ test, ^3^ is nonparametric Wilcoxon rank sum test. ^4^ is Fisher’s exact test.

**Table 7 brainsci-12-00699-t007:** Postoperative complications of 38 patients with cystic pituitary prolactin-secreting macroadenomas.

Variables	Patients (*n* = 38)
Postoperative complications (%)	
Epistaxis	1 (2.6)
CSF rhinorrhea	2 (5.3)
Temporary diabetes insipidus	3 (7.9)
Hypophysis hypofunction	2 (5.3)

## Data Availability

Not applicable.

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
