# Peer review of "Surgical Treatment of Cystic Pituitary Prolactin-Secreting Macroadenomas: A Single Center Study of 42 Patients"

_brainsci, 2022, doi:10.3390/brainsci12060699_

Round 1

Reviewer 1 Report

This is an interesting paper on a single Center experience on  the management of cystic macroprolactinomas. Given the high volume of pituitary neurosurgery, this is a Center of reference, thus this experience can provide important clues on neurosurgical remission of these tumors

I have just few concerns:

Can the authors specify how patients were candidate for medical therapy instead of surgery? Was it a patient's preference?

Furthermore, the tumor volume was slightly lower in the direct surgery group (although not statistically significant) so this might bias the result. Please comment on it

In the Patients section is not really clear where these 12 and 16 patients came from. Furthermore the direct surgery group account 26 cases not 16 if I am correct. Please clarify

Can you please provide the initial dose of bromocriptine for your patients? Can the adverse events reported be due to the initial bromocriptine dose?

Can you specify what kind of "anterior pituitary hypofunction" did you diagnose after surgery? Did it involve only one axis or more?

Discussion results quite long, I suggest to be more concise cutting for instance the part on surgical outcome in prolactinomas

Authors said "surgical treatment should be considered the first choice of therapy for patients with cystic pituitary prolactin macroadenomas"; this might be reasonable on the basis of their results as cabergoline is not available in their Country. This should be specified

Author Response

请看附件。

Reviewer 2 Report

This is an interesting study comparing the clinical outcomes of patients with cystic prolactinomas treated by medication plus surgery or by surgery alone.  The study was well designed, and the findings are significant.

My major concern is that the cases included in this study were not clearly described.  This causes difficulty and confusion in understanding the context fully.  I would suggest a detailed description of each patient group and treatment paradigm.  A comprehensive table describing the clinical features, treatment, and outcome for each patient will be very helpful.

Minor concerns:

  1. “Prolactin macroadenomas” or “prolactin adenomas” should be changed to “prolactin-secreting macroadenomas” or “prolactin-secreting adenomas.
  2. For the two patient groups, it is better to describe them as medication plus surgery group and surgery alone group.
  3. Page 3, Line 124: It is stated that DAs (mostly bromocriptine…). Does it mean a few patients received DAs other than bromocriptine, such as cabergoline?  If so, what was the medications used?  Any different outcomes for these patients?
  4. Page 5, Line 164: Please delete “is”.
  5. Page 5, Line 169: Please delete “S3333”.

Round 2

Reviewer 1 Report

I am satisfied with the novel version of the paper

Reviewer 2 Report

Thanks for the responses.  I have two comments:

  1. Please add the following information to the Patient section in Material Methods:  "In our study, a total of 42 patients were enrolled, of whom 16 patients preferred medical therapy and 26 preferred surgery. Of the 16 patients treated with the preferred medical therapy, four received long-term medical therapy, and 12 eventually underwent surgery. "
  2. I would still like to suggest a main table containing information for each patient as I previously suggested.  The major point is not comparing groups (although a clearly presented table will help the Readers to understand the outcomes of each group), but to provide adequate information about each patient, therefore it will help the Readers to better understand the results so they can make an informed judgement in their own clinical practice.  This table will greatly increase the depth of this manuscript.
